# Sterol 14α-Demethylase Ligand-Binding Pocket-Mediated Acquired and Intrinsic Azole Resistance in Fungal Pathogens

**DOI:** 10.3390/jof7010001

**Published:** 2020-12-22

**Authors:** Katharina Rosam, Brian C. Monk, Michaela Lackner

**Affiliations:** 1Institute of Hygiene and Medical Microbiology, Medical University of Innsbruck, Schöpfstrasse 41, 6020 Innsbruck, Austria; Katharina.Rosam@i-med.ac.at; 2Sir John Walsh Research Institute and Department of Oral Biology, Faculty of Dentistry, University of Otago, PO Box 56, 9054 Dunedin, New Zealand; brian.monk@otago.ac.nz

**Keywords:** sterol 14α-demethylase, lanosterol 14α-demethylase, antifungal resistance, cytochrome P450, azole drugs, azole agrochemicals, heme-containing active site, imidazole, triazole, substrate entry channel, water-mediated hydrogen bond network, treatment, therapy, prophylaxis, point mutations, amino acid substitution, promotor region, ergosterol biosynthesis, tandem repeat, cross-kingdom pathogens, pan-fungal kingdom mutation

## Abstract

The fungal cytochrome P450 enzyme sterol 14α-demethylase (SDM) is a key enzyme in the ergosterol biosynthesis pathway. The binding of azoles to the active site of SDM results in a depletion of ergosterol, the accumulation of toxic intermediates and growth inhibition. The prevalence of azole-resistant strains and fungi is increasing in both agriculture and medicine. This can lead to major yield loss during food production and therapeutic failure in medical settings. Diverse mechanisms are responsible for azole resistance. They include amino acid (AA) substitutions in SDM and overexpression of SDM and/or efflux pumps. This review considers AA affecting the ligand-binding pocket of SDMs with a primary focus on substitutions that affect interactions between the active site and the substrate and inhibitory ligands. Some of these interactions are particularly important for the binding of short-tailed azoles (e.g., voriconazole). We highlight the occurrence throughout the fungal kingdom of some key AA substitutions. Elucidation of the role of these AAs and their substitutions may assist drug design in overcoming some common forms of innate and acquired azole resistance.

## 1. Introduction

The azole drug segment has dominated the global antifungal drug market since the first-generation azoles clotrimazole and miconazole were introduced into the clinic in 1960 [1]. Currently the imidazole and triazole antifungals are the two main classes of azoles used in medicine (Appendix A). The first-generation azoles were imidazoles, while the second and third generation were triazoles [2]. The triazoles antifungals all contain a five-membered di-unsaturated ring system that includes three nitrogen atoms in a heterocyclic core [3] (see Appendix A). The first triazole agents were marketed for agricultural use in 1973, with 31 different agents licensed since. Triazole antifungals were first licensed for use in humans in the 1990s. Today, five triazole drugs are available for therapy of systemic fungal infections: fluconazole (FLC), itraconazole (ITC), voriconazole (VCZ), posaconazole (PCZ), and isavuconazole (IVU) [4,5]. The azoles are used extensively because they are significantly less expensive than other antifungal treatments and their overall antimycotic activity is relatively broad spectrum [6]. Azole drugs are used in: (1) treatment and prophylaxis of livestock (e.g., ringworm in cattle, aspergillosis in poultry, oomycete infections in aquaculture [7,8,9,10], (2) treatment in veterinary medicine (e.g., pets and zoo animals) [11,12], (3) crop protection (e.g., mildews and rusts of grains, fruits, vegetables), (4) plant protection (e.g., leaf spot on flowers, scrubs and trees, including the timber industry) [4], (5) protection of buildings (e.g., antifouling coatings) [13], (f) cosmetic and hygiene products [14], and (6) prophylaxis and treatment in medicine [15].

With fungal pathogens estimated to cause yield losses of up to 30%, the use of fungicides is essential for efficient food production and overall food security [5]. Because azole drugs are amongst the best-tolerated and most active antifungals, they are often the first line treatment in human and veterinary medicine for many fungal diseases [4,16]. Their popularity and extensive use comes at a cost due to the resultant positive selection pressure for azole-resistance amongst fungal pathogens of plants, animals, and humans [17]. An increased incidence of intrinsically azole-resistant fungi and the emergence azole-resistant strains from naturally susceptible fungi has resulted in major losses in food production and in therapeutic failures in both veterinary and human medicine [5]. Many pathogenic molds of humans are saprophytes that decay organic matter and are ubiquitous in the environment, including agricultural soils. Application of the same drug class in medicine and agriculture, together with structural similarities among azole agents, has enabled pathogenic fungi affecting humans to have acquired azole resistance in the environment [4,18]. Moreover, mucormycetes, *Fusarium* spp. and *Aspergillus* spp. are cross-kingdom pathogens that cause infections in plants, animals and humans [19].

The target of antifungal agrochemicals (sterol demethylase inhibitors, SDIs) and antifungal drugs is the cytochrome p450 enzyme sterol 14α-demethylase CYP51 (synonym in molds; e.g., *Aspergillus*), ERG11 (synonym in yeasts; e.g., *Candida* spp.), or SDM (sterol 14α-demethylase; general term), a key enzyme in the fungal-specific ergosterol biosynthesis pathway. It catalyzes the three-step conversation of lanosterol or eburicol to a 14α-demethylated product [20,21,22]. Azole antifungals compete with the substrate by binding to the heme iron in the active site as well as other features in a ligand-binding pocket. The effective binding of azole inhibitors by SDM leads to a depletion of ergosterol, which reduces membrane fluidity, and the integrity in the lipid layer is reduced. Toxic intermediates and sterol precursors such as 14-methyl-3,6-diols accumulate. The consequence is an inhibition of the fungal growth. Azole antifungals tend to be fungistatic rather than fungicidal. This property provides fungi opportunity to adapt to this antifungal class [22,23]. Long-term use, repeated use, overuse and/or under dosing of azole drugs favor the development of azole resistance [24,25,26].

Different mechanisms at the genetic and transcriptional levels allow fungi to become resistant to azole antifungals. The most common of these resistance mechanisms are: (1) amino acid (AA) substitutions in the SDM, affecting the active site of the enzyme, (2) overexpression of the SDM, and (3) overexpression of efflux pumps such as ATP-binding cassette (ABC) transporters and/or major facilitator superfamily (MFS) transporters [22,25,27].

This review will focus on key AA substitutions associated with the ligand-binding pocket of SDMs. We will summarize the known impact of these residues on resistance to short- and long-tailed azoles in pathogenic fungi of humans, animals, and plants.

## 2. Azole Resistance in Fungal Pathogens

### 2.1. Structure of the SDM Ligand-Binding Pocket

Fungal SDM is member of the CY51 family of the cytochrome P450 superfamily found in all eukaryotes. SDM is a bitopic endoplasmic reticulum protein, which spans the lipid bilayer once [20,21,28]. In fungi, it is a key enzyme in the synthesis of the fungal specific sterol ergosterol. In plants, the enzyme is involved in the synthesis of phytosterols (sitosterol, campesterol, and stigmasterol), while in humans and animals the corresponding sterol product is cholesterol [29].

In the yeast ergosterol biosynthetic pathway, the SDM lanosterol 14α-demethylase catalyzes three reduction steps. The 14α-methyl group of the substrate lanosterol is converted first into an alcohol, an aldehyde and then released as formaldehyde together with the insertion of a 14–15 double bond yielding the reaction product 4,4 dimethylcholesta-8(9), 14, 24-trien-3beta-ol for further processing [23]. The enzyme contains an N-terminal membrane associated helix (MH1) and a transmembrane helix (TMH1), which are at an angle of 60° to each other. Hydrogen bond interactions of MH1 with the catalytic domain, which consists of about 450 amino acids, ensure that the catalytic domain is partly embedded in the lipid layer and able to access its substrate (Figure 1) [23,30]. The catalytic domain harbors an iron-containing porphyrin ring, in which the heme receives electrons from a cognate NADPH-cytochrome P450 reductase. The buried catalytic site of the protein is part of a ligand-binding pocket that includes the substrate entry channel (SEC) and a putative product exit channel [23]. Figure 1, Table 1, and Appendix A indicate several AAs important for the binding of the SDM substrate lanosterol and inhibitory agents, respectively

Some AAs involved in azole binding in the ligand-binding pocket and/or involved in azole resistance include G73, Y140, K151, T322, and G464 (AA position according to *Saccharomyces cerevisiae* SDM) (Figure 1). These AAs, together with a range of additional hydrophobic interactions, stabilize ligands in the active center and complement interaction with the heme iron.

Glycine 73 (G73) in *Saccharomyces cerevisiae* is a non-polar amino acid in helix A’ at the mouth of the substrate entry channel. It interacts with distinct parts of long-tailed substrates such as itraconazole (ITC) and posaconazole (PCZ) and results in resistance to these long-tailed antifungals [57,58].

Tyrosine 140 (Y140) in the B-C loop of *S. cerevisiae* SDM structurally aligns with Y132 in *Candida* species (e.g., *C. albicans*, *C. glabrata*, *C. parapsilosis*, *C. auris)*, Y136 in *Ajellomyces capsulatus* (=*Histoplasma capsulatum*) *and Uncinula necator*, Y145 in *Cryptococcus neoformans*, Y121 *in Aspergillus species* (e.g., *A. fumigatus*), and F129 in *Rhizopus arrhizus, Mucor circinelloides*, and *Rhizopus microsporus* (Table 1) [49,59,60]. The hydroxyl group confers polarity on the tyrosine. The substitution most commonly found is a change to a non-polar phenylalanine (F), with the loss of a hydroxyl group affecting the binding of some azole ligands and obliging a hydrogen bond with the heme ring C propionate [59].

Lysine 151 (K151) in helix C of *S. cerevisiae* SDM structurally aligns with K143 in *Candida* species (e.g., *C. auris*, *C. tropicalis*) (Table 2). The positively charged site chain forms an ionic interaction with the carboxylate group of the heme ring C propionate and loses this interaction by becoming surface exposed when lanosterol is bound. Substitution with an arginine (K151R) disrupts the ionic bond due to the larger guanidinium group [30].

Threonine 322 (T322) in helix I of *S. cerevisiae* SDM structurally aligns with T315 in *Candida albicans* and in other fluconazole-susceptible fungi (Table 2). The *A. fumigatus* CYP51A isoform carries a naturally occurring substitution at the structurally aligned position I301. The I301 appears to cause fluconazole resistance in *A. fumigatus*. This was confirmed using an *A. fumigatus* CYP51A knock out mutant and in a CYP51A mutant in which the non-polar I was replaced with the polar threonine (T) residue. Both *A. fumigatus* mutant strains showed lower resistance (20 µg/mL) compared to *Aspergillus fumigatus* strains carrying the naturally occurring I301 polymorphism (640 µg/mL) [36].

In the fungus-specific loop of *S. cerevisiae* SDM, the nonpolar G464 structurally aligns with G448 in *A. lentulus*, G464 in *Sc. apiospermum,* and G464 *Candida albicans*. It is most often substituted with serine (S). The S side chain contains a polar hydroxyl group that may cause azole resistance by modifying the heme environment.

In the following sections, AA substitutions in the above amino acids are discussed in the light of their impact on azole susceptibility. An overview will be provided for various pathogenic fungi of plants and animals (Table 1). AA positions were reported according to their position in the *S. cerevisiae* SDM.

### 2.2. G73 in S. cerevisiae SDM

Some AA substitutions have been reported at glycine 73 (G73). The structurally aligned mutations G54E/R/W in *A. fumigatus* CYP51A confer resistance to ITC and PCZ, but retain susceptibility to VCZ. This is thought to be due to long-tailed azole-specific interactions with the mouth of the SEC [61,62]. Sagatova et al. generated *S. cerevisiae* mutant strains carrying the following SDM mutations: G73E, G73R, and G73W, respectively [58]. The SDM G73E/R enzymes supported yeast growth, but SDM G73W did not, suggesting a significant impediment in the SEC. The SDM73E/R strains showed 2.5-fold increases in susceptibility to FLC, VCZ, and ITC and type II binding studies showed these drugs and PCZ bound tightly to the affinity purified wild-type and mutant enzymes. Crystal structures obtained for the SDM Y73E and SDM 73W enzymes showed that ITC adopted bent conformations different to that found with the wild-type enzyme [58]. Despite visualization of inhibitor conformation, the inability of the mutated yeast SDM to mimic the expected resistance pattern reported by Alcazar-Fuoli et al. for *A. fumigatus* CYP51A expressed in *S. cerevisiae* is an important limitation that emphasizes the need to obtain a full-length structure of *A. fumigatus* CYP51A [63].

### 2.3. Y140 in S. cerevisiae SDM

Mutations or substitutions structurally aligned with SDM Y140F are prevalent in many species, e.g., *A. fumigatus* CYP51A Y121F, *Candida* species SDM Y132F, *Cryptococcus neoformans* SDM Y145F, *Scedosporium apiospermum* SDM Y136F, *Ajellomyces capsulatus* SDM Y136F, and mucormycete SDM Y129F. The same modification is frequently found in fungal phytopathogens, e.g., *Blumeria graminis* SDM Y136F and *Parastagonospora nodorum* SDM Y144F (Table 1).

The Y140F substitution is important as the Y hydroxyl group not only forms a hydrogen bond with the heme ring C propionate but can also form a water-mediated hydrogen bond with the tertiary alcohol of short-tailed azoles such as FLC and VCZ (Figure 1). The Y140F mutation therefore leads to a lower affinity for such drugs and may also reduce the stability of the enzyme [59]. Some known resistance phenotypes are summarized in Table 1. Fungal pathogens of plants and animals harboring this AA substitution are resistant to commonly used short-tailed azoles. Positive selection pressure in azole-rich environments leads to increased prevalence of isolates with SDM mutations equivalent to Y140F [64,65,66]. Some intrinsically resistant species (e.g., mucormycetes) also carry the comparable substitution [49]. In agriculture, SDIs such as tebuconazole (TBC), epoxiconazole (EPC), and triadimenol (TDM) are used. The structures of some SDI and azole drugs used most commonly in agriculture and medicine are given for comparison in Appendix A. There are significant structural and functional similarities between FLC and VCZ and the SDIs used in agriculture. The agrochemicals are of comparable size to FLC and VCZ and occupy the active site, but not the SEC. Like FLC and VCZ, the agrochemicals TDM, TBC, and prothioconazole-desthio all contain a hydroxyl group capable of water-mediated hydrogen bond contact with residues equivalent to *S. cerevisiae* SDM Y140. Recombinant *S. cerevisiae* SDM Y140F/H mutants confer significant resistance to *R*-TBC and *S*-prothioconazole-desthio [67]. These two compounds show tight type II binding to *S. cerevisiae* SDM and in crystal structures are in positions that would allow for the formation of a water-mediated hydrogen bond network with Y140. In contrast, no significant resistance was conferred to prochloraz, which cannot form such as network and is more weakly bound in the active site of *S. cerevisiae* SDM despite being an imidazole rather than a triazole. Surprisingly, the *S. cerevisiae* SDM Y140F/H mutations conferred significant azole resistance to the triazole agrochemical difenconazole which contains a 4-methyl-1,3 dioxylane ring that was shown to be in crystal structures to displace the water contributing to Y140-associated hydrogen bond network [67]. Its resistance might instead be due to effects on the Y140F/H mutations on the alignment of the dichlorophenoxyphenyl head group of difenoconazole with the F384 in helix I.

The binding of long-tailed azoles within the active site also involves the heme iron and hydrophobic interactions in common with the short-tailed azoles, but the binding of ITC and PCZ is unaffected by the Y140F/H substitution as these drugs do not contain the relevant tertiary alcohol group. The long-tailed azoles also have additional interactions within the SEC that are not available to the short-tailed azoles bound within the active site. For example, the N1 in the piperazine ring of itraconazole and posaconazole interacts with a water-mediated hydrogen bond network involving the main chain amino groups of H381 and S382 in *S. cerevisiae* SDM (Figure 1). This water-bearing pocket is probably important for locating the hydroxyl group so that lanosterol binds in a catalytically competent position in the active site (Figure 2).

#### 2.3.1. *Aspergillus* Species

The genus *Aspergillus* includes several pathogenic species of which *A. fumigatus* is the most abundant [25]. Aspergilli can cause allergies and both acute and chronic infections [65]. The most recent ESCMID (European Society of Clinical Microbiology and Infectious Diseases) -ECMM (European Confederation of Medical Mycology)-ERS (European Respiratory Society) guideline recommends VCZ as first line treatment of invasive aspergillosis, with liposomal amphotericin B (AmB) or isavuconazole (IVU) as alternative therapies [68]. VCZ is the treatment of first choice for aspergillosis due to its broad-spectrum activity that also covers infections with less common aspergilli such as *A. terreus*, its cost advantage compared with liposomal amphotericin B, the availability of oral and intravenous formulations, and its better tolerability than AmB [69,70].

Aspergillosis not only affects humans but also poultry [71], captive wild birds (e.g., water birds and penguins in zoos) [72] and wild birds [73]. Recently, an outbreak of aspergillosis was reported in the endangered kakapo population of New Zealand [74]. The same range of antifungals used in human medicine is used to treat avian aspergillosis [73]. Aspergilli are saprophytic fungi ubiquitous in the environment (including soil and compost). They are dispersed widely by air and are among the most commonly detected airborne fungi [75]. Some industries, such as tulip production, apply large quantities of azole agrochemicals. The abundance of azole-resistant *A. fumigatus* strains has been correlated with close proximity to azole-rich environments [76,77,78].

The rise of VCZ-resistant isolates of *A. fumigatus* has reduced the efficacy of VCZ prophylaxis and treatment. While the *A. fumigatus* CYP51A TR34 L98H mutant was initially reported, CYP51A TR46/Y121F/T289A is among the most prevalent mutations found in clinical and environmental isolates of this pathogen [79,80]. Y121F in *A. fumigatus* CYP51A predominantly occurs together with the T289A mutation in helix I plus the TR46 (46 base pair tandem repeat) in the CYP51 promoter. Although, T289A occurs in natural isolates, but does not itself confer azole resistance [31]. Snelders et al. used laboratory strains to show that TR46 results in overexpression of the *CYP51A* gene and increased production of SDM [31]. They also found that while the TR46/Y121F mutant showed high voriconazole and itraconazole resistance (>8.0 mg/L), the TR46/Y121F/T289A mutant was susceptible to itraconazole (0.5 µg/mL) but not to voriconazole (>8 µg/mL). It was suggested that T289A mutation moderates resistance to itraconazole but is needed to counterbalance the harmful effect of Y121F on normal protein function. This view is supported by the fact that the three mutations are most frequently found together in environmental isolates [31]. The abundance of strains with the resistance genotype TR46/Y121F/T289A is increasing, especially in environments with positive selection pressure for azole resistance (e.g., in vineyards and tulip compost) [77,81].

VCZ resistance is not restricted to *A. fumigatus sensu stricto* and occurs in other species of the section *Fumigati.* The *Fumigati* contains closely related species that are morphologically indistinguishable from *A*. *fumigatus* [82]. Among sibling species pathogenic in humans are *A. lentulus* and *A. felis* [83]. Reduced azole susceptibility is a species-specific feature of *A. lentulus,* although the exact mechanism has yet to be elucidated. Minimal inhibitory concentration (MIC) levels are generally high for ITC (0.43–16 μg/mL), VCZ (3–7.5 μg/mL), and PCZ (0.12–2.0 μg/mL) [84,85]. Experimentally, this reduced susceptibility to azole drugs (5- to 100-fold) can be attributed to the *CYP51A* gene. Deletion mutants of *CYP51A* (Δ*cyp51*A) had lower MICs to ITC (0.03–0.06 μg/mL), VCZ (0.25 μg/mL), and PCZ (0.015 μg/mL). *A. fumigatus* and *S. cerevisiae* mutants, carrying the *A. lentulus CYP51A* gene, showed higher MICs to ITC (0.5–8.0 μg/mL), VCZ (1.0–4.0 μg/mL), and PCZ (0.12–0.25 μg/mL) [86]. Alcazar-Fuoli et al. used a three-dimensional model of *A. lentulus* CYP51A to suggest that the lower susceptibility to VCZ of *A. lentulus* is due to decreased affinity between the azole and the SDM [37]. The rare, but potentially underdiagnosed species *A. felis* has also a reduced susceptibility to VCZ and ITC, but the mechanism responsible has not been determined. The possibility of a Y121F mutation was suspected, but this idea requires further research [86].

#### 2.3.2. *Candida* Species

*Candida* species are the most frequently diagnosed fungal pathogens in humans. They can cause a variety of diseases, ranging from superficial mucosal infections to deep-seated infections such as candidemia [16]. The most prevalent species is *C. albicans*, followed by *C. glabrata*, *C. tropicalis*, *C. krusei*, and *C. parapsilosis* [16,87]. A species of increasing concern is *C. auris*. This species has recently evolved as a significant human pathogen. It was first isolated in 2007 as an azole-susceptible strain from an aural sample of a female Japanese patient [60]. The first hospital outbreak was reported in Europe in 2016 and was due to a multidrug-resistant strain of the species *C. auris* [88]. Since then, *C. auris* strains have spread globally and they are now recognized by the CDC to be a serious global health threat [89].

Globally, the antifungal agent most commonly used for the treatment of *Candida* infections is fluconazole, followed by the echinocandins [90]. Fluconazole resistance in *Candida* species is a major concern. The majority of the *Candida* species, including *C. albicans*, are naturally fully susceptible to fluconazole. This contrasts with molds, which are all intrinsically resistant to fluconazole. Resistance against FLC and VCZ in *Candida* species has frequently been linked to the SDM Y132F/H mutations. These SDM mutations have been reported for species including *C. albicans*, *C. tropicalis,* and *C. parapsilosis* and they lead to both FLC and VCZ resistance. (Table 1). The SDM Y132F substitution has been most extensively studied in *C. albicans*. It significantly (16-fold) increases VCZ and FLC MICs in this yeast while the Y132H gives an 8-fold increase [45].

*Candida parapsilosis* has emerged as the most prevalent *Candida* species detected in South Africa (58–63%). In other regions, including Latin America, the United States of America, Asia, and Western Pacific, the prevalence of *C. parapsilosis* is much lower (1%–7.7%). A 2009–2010 study showed that approximately 80% of all *C. parapsilosis* strains are FLC resistant. The SDM Y132F mutant was found in 68% of all FLC-resistant strains [27]. The incidence of FLC resistance in *C. parapsilosis* is alarming as FLC is the antifungal drug used most often in developing countries [91]. *C. parapsilosis* is a commensal of the skin and, unlike other fungi, is easily transmitted by healthcare workers, resulting in major hospital outbreaks. Isolates causing such outbreaks tend to be more virulent than isolates causing sporadic cases [92]. In Turkey, a single center study tracked the prevalence of *C. parapsilosis* with the evolution of azole resistance over time. The FLC resistance can either be acquired due to the positive selective pressure in azole-rich environments or due to horizontal transfer from other individuals. Arastehfar et al. found that SDM Y132F and the SDM Y132F K143R combination occurred in clinical outbreak strains [91]. The outbreak strains were of clonal origin and this correlated with the abundance of strains harboring SDM Y132F. Mortality rates of patients infected with FLC-resistant *C. parapsilosis* strains were higher than for patients infected with FLC-susceptible strains [91].

A major concern is the multidrug resistance of *Candida auris* isolates. In the last five years, four phylogenetically clades of *C. auris* have been described that correlate with different geographical origins: the South Asian clade, South American clade, African clade, and the East Asian clade [93]. Each clade has specific resistance profiles and carries different adaptive mutations [94]. The SDM F126T mutation is strongly associated with the African clade while SDM Y132F is most prevalent in the South American clade. Strains of the South Asian clade carry most frequently the Y132F and/or K143R substitutions, especially clinical isolates from India and Pakistan [94,95]. As the type strain (East Asian clade) isolated in 2007 is fully susceptible to all systemically applied antifungal agents, *Candida auris* has revealed how antifungal resistance can evolve quickly in an emerging fungal pathogen. In less than two decades, and depending on the clade, up to 100% of *Candida auris* strains have become FLC resistant and 41.1% VCZ resistant [60,96,97]. The two most common mutations found in *C. auris* SDM are Y132F and K143R. The latter mutation will be discussed in Section 2.3.4. The haploid nature of *C. auris* allows for such a rapid resistance development.

#### 2.3.3. *Cryptococcus* Species

The genus *Cryptococcus* includes several human pathogens. The most common of these are *Cr. neoformans* and *Cr. gattii*. *Cryptococcus neoformans* causes meningoencephalitis, mostly in immunocompromised patients, such as AIDS patients [44,98]. Cryptococcosis contributes significantly to the high mortality rates of HIV-infected patients worldwide. In addition, some immunocompetent patients can suffer from cryptococcosis [44,98]. *Cryptococcus* grows as yeast in the host and the environment. In the human host, the fungus grows as a capsulated yeast. Similar to *Candida*, *Cryptococcus* is naturally fully susceptible to fluconazole. Fluconazole as a single drug therapy or combination with 5-fluorocytosine (5-FC) is a first line treatment for cryptococcosis [99]. Fluconazole is often the treatment of choice as the drug is inexpensive, widely available in emerging economies and can be used for long-term therapy due to its low toxicity and good penetration of the central nervous system [44].

*Cryptococci* are basidiomycetes that cause major disease outbreaks around the globe. As these outbreaks are clonal, the increased incidence of fluconazole resistance in *Cryptococcus* populations is a major concern [100,101,102]. *Cryptococcus neoformans* and *Cr. gattii* have innate heteroresistance to azoles, an adaptive mechanism involving duplication of chromosomes containing specific genes that confer azole tolerance [103].

The *Cyptococcus* SDM Y145F mutation (SDM Y140F in *S. cerevisiae*) is one of the mutations that confers resistance to FLC and VCZ. A biochemical test of this relationship is needed for *Cr. neoformans* and *Cr. gattii* [44]. In Taiwan, the frequency of resistance to FLC among *Cr. neoformans* strains has increased from 0–33% in 2001–2006 to 75–88% in 2011–2012 [104]. In order to track and control resistant *Cryptococcus* strains, MIC determinations should be standard practice [105].

#### 2.3.4. *Saccharomyces cerevisiae*

*S. cerevisiae* infections in humans are very rare. However, blood stream infections due to *S. cerevisiae* have been reported in COVID-19 patients [106] In the West, the yeast *Saccharomyces cerevisiae* is the fungus used most in food industries, e.g., fermented foods including wine, bread, chocolate, and beer [106]. *S. cerevisiae* is also used to produce biofuel [107], dietary supplements [108,109], and it provides cell factories that produce human insulin and other medications [110].

In the life sciences, *S. cerevisiae* is one of the most widely used eukaryotic model organisms because it is easy to culture, genetically manipulate and study biochemically [111]. *S. cerevisiae* has been used as a model to elucidate the impact of SDM ligand-binding site mutations in various pathogenic fungi [59]. In order to assess azole susceptibilities of heterologous expressed SDMs, a host strain deleted of drug efflux pumps can be used to minimize their effect. In addition, if the heterologous expressed SDM is functional, the native SDM can be deleted [59]. The first crystal structures of full-length heterologous expressed recombinant fungal SDM in *S. cerevisiae* were from *S. cerevisiae* itself and then from *C. albicans* and *C. glabrata*. The high-resolution structures of wild-type and mutant SDMs in complex with substrates and inhibitors provide insight into associated phenotypes and biochemical changes and can now underpin structure-directed antifungal research and discovery [20]. Thus far, *S. cerevisiae* SDM has provided an effective surrogate to study mutations in the SDM of *Candida* species [21,58].

#### 2.3.5. *Scedosporium* species

Azole resistance in *Scedosporium apiospermum* is not well understood. To our knowledge, a single study has analyzed azole resistance patterns and correlated them with AA substitutions in the *Sc. apiospermum* SDM [50]. Similar to other molds, *Scedosporium* species have two SDM-encoding homologs, namely, *CYP51A* and *CYP51B*. VCZ is the first line therapy for scedosporiosis [112]. The susceptibility of *Scedosporium* to azole drugs seems to be species specific. *Sc. aurantiacum* strains are often resistant to PCZ while the majority of strains in the *Sc. apiospermum* species complex are susceptible to PCZ, and with PCZ resistance more common than VCZ resistance [113]. Bernhardt et al. described VCZ-resistant *Sc. apiospermum* strains harboring the SDM Y136F substitution in their *CYP51*A gene [50]. While this mutation in the SDM active site may be prevalent in *Scedosporium* species, further research is needed to elucidate its impact on resistance to short-tailed azoles.

#### 2.3.6. *Ajellomyces capsulatum*

The dimorphic ascomycete *Ajellomyces capsulatum*, better known in medical settings as *Histoplasma capsulatum,* is the causative agent of histoplasmosis, the most frequent endemic mycosis in the United States of America [114]. The dimorphic fungus grows as a filamentous fungus in the environment and as a yeast in the human host. It is a progressive clinical illness that can spread from the lung, especially in immunocompromised patients [115]. ITC is the azole used most frequently to treat histoplasmosis although FLC is an option when ITC cannot be applied for other reasons [114]. One third of histoplamosis patients treated with FLC experience a relapse. Isolates from relapse patients were found to have higher FLC MICs [114]. Their FLC resistance was linked to SDM Y136F and these isolates had a 4-fold higher VCZ MIC [114].

#### 2.3.7. Innate Azole Resistance of Mucormycetes

The incidence of mucormycete infections appears to be increasing globally. Their incidence is under-reported as the disease is often misidentified because it is often hard to differentiate from other mold infections. Several mucormycetes cause severe infections in immunocompromised patients. The prevalence of individual species varies between countries. In Europe and Africa, the species reported most frequently as causing mucormycosis was *Rhizopus arrhizus* followed by *Lichtheimia corymbifera* [116]. Patients suffering from invasive mucormycosis have mortality rates that approach 96%. The current first line treatment is liposomal amphotericin B, with posaconazole used as a salvage therapy. Mucormycetes are intrinsically resistant to short-tailed azoles. VCZ, used widely as prophylaxis against aspergillosis in high-risk patients is not effective for mucormycosis. This intrinsic resistance involves two natural occurring SDM substitutions, F129 and V291 [49]. While other fungi can acquire comparable resistance-conferring mutations, the substitutions in mucormycetes seem to be evolutionary conserved. Thus far, they have been found in all mucormycete whole genome sequences.

#### 2.3.8. Agricultural Pathogens

The triazoles are amongst the most widely used fungicides in agriculture. The first SDM Y136F mutation found in a fungal phytopathogen was associated with reduced TDM susceptibility. SDM Y136F has now been reported in a variety of agricultural important pathogens*: Blumeria graminis* (Y136F), *Mycosphaerella graminicola* (Y137F), *Parastagnospora nodorum* (Y144F/H), *Puccinia triticina* (Y134F), and *Uncinula necator* (Y136F) (Table 1 and Table 2). The associated azole resistance patterns are shown in Table 1. The chemical structures of the triazoles used in agriculture and medicine are often similar (Appendix A), so the occurrence of comparable mechanisms of resistance among crop and medically important pathogens is not surprising [117]. According to the risk assessment of the European center for disease control and prevention (ECDC), the most used triazoles in agriculture are prothioconazole, TBC and EPC in the United Kingdom, Denmark, and the Netherlands. Chemically, they belong to the group of short-tailed azoles [118] (Appendix A).

*Mycosphaerella graminicola* (*Zymoseptoria tritici*) causes septoria leaf blotch in wheat, which causes crop losses of up to 50% [43]. SDIs are fungicides commonly used against the septoria leaf blotch. Exposure of *Mycosphaerella graminicola* to TDM from the 1980s led to the acquisition of the SDM Y137F mutation. Its replacement with EPC and prothioconazole has resulted in the emergence of a range of alternative SDM mutations [52,53].

*Blumeria graminis,* the most common ascomycetous fungal pathogen infecting barley crops, causes a powdery mildew that reduces barley yields by 20–40% [51,119]. In *Blumeria graminis*, SDM Y136F has been found together with the S509T mutation [51]. Isolates with this genotype are resistant to TBC [51].

*Parastagonospora nodorum* is an ascomycete and the causative agent of the global crop disease *Stagnospora* or *Septoria nodorum* leaf and glume blotch [54]. The economic impact of this fungal disease is difficult to determine, but has been estimated to cause up to 30% yield loss in Australia. Isolates of *Parastagonospora nodorum* with a reduced susceptibility to propiconazole or TBC were found to carry either the SDM Y144F or Y144H mutations, respectively [54].

*Puccinia triticina* is a basidiomycete that causes the brown rust or leaf rust. It is a common pathogen of wheat that causes up to 50% crop loss [120]. The acquired SDM Y134F mutation has been reported to contribute to reduced EPC susceptibility, but additional mechanisms may contribute to the resistant phenotype [121]. An accompanying mutation similar to *A. fumigatus* T289A may be involved.

*Uncinula necator* is an ascomycete that causes powdery mildew. It is the major fungal pathogen found in grapevines. Powdery mildew is controlled by extensive use of fungicides, including SDIs [121]. Its economic importance stimulated the first study of the mechanism underlying azole 2resistance in a crop pathogen. Modification in the SDM ligand-binding pocket has been linked with SDI resistance patterns [56]. SDM Y136F was found to be associated with TDM resistance [122], with isolates carrying this point mutation having reduced susceptibility to TDM [123].

The impact of azole-resistant fungal pathogens in agriculture is an underrated problem. Most crops needed for the food supply are affected by fungal pathogens. Harvest losses caused by phytopathogens can be expected to cause food shortages for a steadily increasing human population. With extensive use of SDIs, resistance patterns that compromise their efficacy will continue to evolve. Novel antifungal agents are needed in human medicine and to guarantee efficient food production.

Further research is needed to understand why the mutations structurally aligned with *S. cerevisiae* SDM Y140F/H are, or have been, prevalent in fungal pathogens of medical or agricultural importance. We hypothesize that it represents a pan-fungal kingdom mutation that is readily induced by small azole antifungals reliant on interactions within the active site of SDM.

### 2.4. K151 (According to S. cerevisiae SDM)

The SDM K143R substitution is an acquired mutation in *Candida* species, including *C. auris* and *C. tropicalis* [43,124]. It has been suggested that the SDM K143R mutation interferes with the entry of the azole or its binding with the active site [97]. The substitution is predicted to modify the ionic bond with the heme ring C propionate and affect the conformation of β-bulge [20].

The high FLC and VCZ resistance of the emergent multidrug-resistant pathogen *C. auris* has been associated with a K143R mutation that confers MIC ranging from 32 to 128 µg/mL for FLC and 0.25 to 2 µg/mL for VCZ [125,126].

There have been several outbreaks of *C. auris* in hospital settings, especially in intensive care units, e.g., at Oxford University Hospital. Transmission was shown to be due to reusable axillary temperature probes. The testing of isolates according to the Sensititre YeastOne system found that 100% of the strains were resistant to FLC, 98% to VCZ, and 90% to PCZ, according to *C. albicans* breakpoints [127]. Other recent outbreaks have occurred in Valencia [128] and London [88].

Heterologous expression of in *Escherichia coli* of *C. albicans* SDM carrying the K143R mutation was reported to confer fluconazole resistance in vitro [97].

### 2.5. T322 (According to S. cerevisiae SDM)

#### 2.5.1. *Aspergillus fumigatus*

*A. fumigatus* has two homologous copies of the SDM-encoding genes, namely, *CYP51*A and *CYP51*B*. CYP51A* is considered to be the workhorse of the ergosterol pathway. A recently discovered natural CYP51A I301 polymorphism was postulated to cause the FLC resistance of *A. fumigatus*. In the CYP51B homologue of *A. fumigatus*, T304 is structurally aligned with I301 [36]. The I301 polymorphism results in the loss of a possible hydrogen bond that has been linked to a change in the flexibility of helix I [36]. CYP51A I301 together with S297 have been postulated to be essential for the modified interaction of FLC within the active site of the enzyme [36].

#### 2.5.2. *Candida albicans*

Lamb et al. postulated that the acquired SDM T315A mutation *Candida* species leads to fluconazole resistance [41]. Using *S. cerevisiae* SDM mutants, they found T315A, which is equivalent to I301 in *A. fumigatus* CYP51A (see 2.5.1), reduces fluconazole susceptibility 4–5-fold. As this mutation is due to one single nucleotide polymorphism, it might be acquired easily in environmental and clinical settings [41].

### 2.6. G464 (According to S. cerevisiae SDM)

#### 2.6.1. Glycine G464S/D

The SDM G484S mutation has been linked with FLC resistance [47]. Forastiero et al. also constructed *C. tropicalis* G484D mutants [129]. It was suggested that the *C. tropicalis* SDM G464S or G464D affects the environment of the SDM heme group due to the nucleophilic serine or acidic aspartic acid side chains [129]. *C. albicans* SDM G464S gave a 2- and 4-fold increase in FLC and VCZ MICs, respectively. The combination of SDM G464S and D278N further increased resistance to FLC and VCZ [130].

The use of *S. cerevisiae* models showed that resistance to FLC was conferred by G464H alone or in combination with other mutations. The combinations included Y132H and R467K or Y132H and H283R. The G464S Y132H combination showed a 32-fold increase in FLC resistance. As these amino acids are located at, the opposite site of the heme, crystal structures of *S. cerevisiae* SDM were used to show that the G464S hydroxyl replaces a water that normally creates a hydrogen bond with the heme ring D propionate [58].

*Sc. apiospermum* SDM G464S structurally aligns with the *A. fumigatus* CYP51A G448S. Mutations in G448 were reported to confer reduced susceptibility to VCZ, ITC, and PCZ, but further information on the binding of antifungal agents and its impact on azole resistance is needed [50].

The *Cr. neoformans* SDM G484S structurally aligns with *S. cerevisiae* SDM G464S. Its correlation with the development of azole resistance led to *Cr. neoformans* SDM G484 being suggested as a mutational hot spot [47]. The *Cr. neoformans* SDM G484S conferred FLC, but not to VCZ or ITC resistance [47].

#### 2.6.2. Glycine G460 in *M. graminicola*

*Mycosphaerella graminicola* SDM G460 mutations occur frequently. The *M. graminicola* SDM G460D mutation was first described in 1992. Strains with SDM G460 deletions (Δ460) were also found in Germany in 2004 [131]. *Mycosphaerella graminicola* SDM Δ459 andΔ460 strains have been successfully engineered. These deletions and other AA substitutions in positions 459–461 are prevalent in several environmental strains with increased resistance to azoles. As G460 is part of the fungus-specific loop, early studies suggested it is involved with docking with TDM or prochloraz. The Y459 and G460 deletions were proposed to increase cavity volume and impact on resistance patterns [52]. It more likely that these mutations cause a misfolded fungus-specific loop that affects the binding efficiency of the cognate NADPH-cytochrome P450 reductase. The reductase is essential for reducing the SDM heme iron. Interactions of SDIs with the heme including direct completion with the substrate can be expected to more strongly affect the activity of short-tailed SDIs that are bound entirely within the active site. Such interactions have a lesser effect on long-tailed SDIs that have additional interactions with the SEC.

## 3. Discussion

Mutations or substitutions structurally aligned to *S. cerevisiae* SDM Y140F/H are prevalent in many fungal pathogens of plants, animals, and humans. This type of modification appears to provide a pan-fungal kingdom resistance mechanism for fungi exposed to short-tailed azole drugs or agrochemicals. The mutation is found in basidiomycetes (*Cryptococcus neoformans, Puccinia triticina*) ascomycetes (*Ajellomyces capsulatus, Aspergillus* spp., *Candida* spp*. Kluyveromyces marxianus, Scedosporium apiospermum, Blumeria graminis, Mycosphaerella graminicola, Parastagonospora nodorum, Uncinula necator)* and mucormycetes (*Rhizopus arrhizus, Rhizopus microsporus,* and *Mucor circinelloides*). Contamination of soils and sweet- and salt-water bodies with azole antifungals, due to agricultural leaching and aquaculture makes it likely that comparable substitutions are likely to occur even in phylogenetically ancient groups of fungi such as chytridiomycetes and oomycetes. In ascomycetes and basidiomycetes, mutations equivalent to Y140F/H are acquired adaptations to azole drug exposure [44,132]. The structurally aligned SDM F129 is a naturally occurring substitution that is evolutionary conserved among mucormycetes [49]. The mutation has been linked with resistance to short-tailed azoles (e.g., VCZ, FLC, TBC, PPC, TDM, and EPC) in fungal pathogens of both plants and humans (Table 1). This resistance mechanism indicates the importance that this AA position in the binding short-tailed SDIs. The hydroxyl group of the tyrosine forms a structure—a stabilizing hydrogen bond with the heme ring C propionate and a water-mediated hydrogen bond network with the tertiary hydroxyl group in short-tailed azoles including FLC, VCZ, TDM, and TBZ in the target active site. This binding is important for short-tailed azoles as they form fewer interactions in the ligand-binding pocket than long-tailed azole drugs such as ITC and PCZ, which form additional hydrophobic contacts and water-mediated hydrogen bonds in the SEC. Although both short- and long-tailed azoles bind to the heme iron and have multiple hydrophobic interactions within the active site, for short-tailed azoles, the water-mediated hydrogen bonding in the active site contributes to an overall binding affinity that allows for effective competition with substrates such as lanosterol and eburicol.

In mucormycetes and aspergilli, an accompanying structurally aligned innate substitution (A291 in mucormycetes) or acquired mutation (T289A in *Aspergillus*) was postulated to confer resistance to the short-tailed azole VCZ (Table 1). For mucormycetes, it remains to be proven, experimentally, that the combination of these F129 and A291 substitutions is essential for resistance to both VCZ and FLC. In addition, in *Aspergillus* insertion in the *CYP51A* promoter of the 46 bp, TR46 tandem repeat was also required to confer the resistance [31]. While SDM Y121F alone increased the VCZ MIC to 4 mg/mL and T289A alone did not affect the MIC, both mutations were required to increase the VCZ MIC to 8.0 µg/mL. When TR46 was inserted into the strains carrying either Y121F or both mutations (Y121F and T289A), the MIC for VCZ was >8.0 µg/mL. Isolates carrying either or both active site mutations, but lacking the TR46 promoter modification, remained fully susceptible to ITC and PCZ. These results confirmed that the combination of the two active site mutations plus enzyme overexpression conferred resistance to short-tailed azoles.

Other ligand-binding pocket associated substitutions detected less frequently are known to be involved in acquired or intrinsic resistance but have not been found in such a diverse range of fungal pathogens. SDM K143R/Q mutations have been described as single substitutions or occurring with Y132F. Thus far, this mutation has only been described in *Candida* species [43,122,124]. In particular, non-*albicans* species (*C. auris*, *C. tropicalis*, *C. orthopsilosis*) and strains causing clinical outbreaks have been found to carry such mutations [43,124,125]. In molds such as *Aspergillus*, this residue does not seem to play a role in azole resistance.

Intrinsic fluconazole resistance in *A. fumigatus* appears to be mediated via SDM 301I [36]. This finding was confirmed using recombinant strains. An I301T mutant exhibited a much lower FLC MIC (20 µg/mL) than the parental strain (640 µg/mL) while the MIC for other azole drugs was unchanged. SDM in FLC-susceptible fungi such as *Candida albicans* carries the structurally aligned polar AA T315. This residue is too distant to affect the binding of FLC directly and it sidechain hydrogen bond is instead thought to increase the flexibility of helix I. The non-polar I301 in *A. fumigatus* SDM cannot form this hydrogen bond and will therefore expected to stiffened helix I. Whether this AA substitution mediates FLC resistance in other intrinsically FLC-resistant molds remains to be determined.

Mutations structurally aligned with *S. cerevisiae* SDM G73 and G464 are infrequent. A more comprehensive picture of the relevance of all the ligand binding site substitutions and their impact on long- and short-tailed azole resistance requires further structural and functional analysis.

We are confident that AA mutations/substitutions structurally aligned with *S. cerevisiae* SDM Y140, K151, and V311 we have described represent the tip of an iceberg across the fungal kingdom. Fungi occur in almost all known ecological niches. Most of these niches can be contaminated with azoles, but their impact is poorly understood. While it is clear that there is positive selection pressure for azole-resistant strains and species, how the complex composition of fungal communities in different niches is affected is not known. Current reports of SDM binding site mutations are limited to a few fungal pathogens of plants and animals. The effects of azole contamination that does not directly affect commercial production or human health are not known, e.g., how is fungal biodiversity affected in azole contaminated water and soils?

Fungi with acquired azole resistance are likely to have a fitness advantage in azole-contaminated environments [18,133,134,135,136,137]. In addition, fungi such as mucormycetes that are intrinsically resistant to short-tailed azoles should benefit from a positive selection pressure in azole-rich environments and outcompete azole-susceptible species [137]. It is also problematic that agrochemicals such as TDM have very long half-lives, ranging from 110 to 375 days in soil [138]. Azole residues have been detected in foods including strawberries, apples, grapes, and peppermint, in some cases reaching peak values of up to 0.5 mg/kg to 0.8 mg/kg [139]. It has been shown that agricultural imports provide a possible route for the intercontinental spread of resistant fungi. This raises the concern that strains harboring *CYP51*A mutations may encroach into clinical settings [76,77]. Azole inhibitor intake could favor selection of azole-resistant *Candida* species or isolates with acquired azole-resistance in patients’ mycobiomes. The use of cosmetics and personal hygiene products containing azole antifungals may aid colonization of hair and skin with azole-resistant *Candida* species [140]. While little evidence supports a shift towards an azole-resistant mycobiome in patients due azole contaminated food and hygiene products, previous azole-treatment in patients is clearly linked with azole treatment failure [24].

High prevalence of mucormycetes in homes and workspaces (bakeries, pig farms, taxis, waste-sorting plants) has been confirmed [137]. Selection pressure contributes to changes in the mycobiome composition of such niches. This may partially explain the rise of mucormycete infections worldwide. The use of FLC/echinocandins and VCZ as a first line prophylaxis against the most prevalent fungal pathogens *Candida* and *Aspergillus*, respectively, has created azole-rich niches in patient cohorts. Because mucormycetes are intrinsically resistant to FLC, echinocandins and VCZ, high-risk patients receiving antifungal prophylaxis with any of these three agents can provide a suitable niche for these fungi. The presence of SDIs in agricultural soils and on food might contribute to significant exposure of these patient cohorts. For example, contamination of flour and bakeries with azole-resistant aspergilli and mucormycetes has recently been found and the associated risk of exposure to bakers of these azole-resistant fungi discussed [141]. Professions at risk of even higher exposures include agricultural workers, farmers, composters, gardeners, and millers. Since the majority of mold infections are airborne and azole-resistant molds may be readily dispersed by wind, susceptible individuals living close to farms with intensive azole use are at risk, as has been shown for the Netherlands and the UK [76]. Furthermore, azole-resistant aspergilli associated with tulip bulbs and potted plants can be transported from the flowerbed to the patient bed, e.g., from the Netherlands to distant countries such as Japan [76,77].

We need to limit the use of azole drugs and agrochemicals by applying them more judiciously. A key aim should be to reduce azole contamination of the environment and in food. Ideally, azoles should be used for very specific applications and a strict structural separation of azole inhibitors used in agriculture and medicine is required. The wide availability of azoles favors the emergence of azole-resistant fungi. Within a few decades, azole-resistance may become so wide spread that the utility of this valuable substance class will be lost to both agriculture and medicine. By understanding the molecular basis of azole binding to its target, it should now be possible to design azole inhibitors that meet these challenges.

## 4. Conclusions

New antifungal compounds are urgently needed due to the emergence of azole- and multidrug-resistant fungi. Understanding the mechanism of acquired and intrinsic azole resistance in a wide range of fungal pathogens is therefore highly relevant to the field of antifungal drug discovery.

The linkage of azole resistance in agriculture and medicine is a major problem that needs to be addressed. Two major concerns are: (1) the exposure of people to azole drugs via azole-contaminated food products and subsequently the selection pressure on the human mycobiome, and (2) the selection for and evolution of azole-resistant fungal pathogens in the environment. Azole resistance in the medical setting is particularly troublesome since only three major substances classes are available for the treatment of systemic fungal infections (polyenes, echinocandins, and azoles). Some fungal pathogens of humans have intrinsic resistance against one or multiple drug classes, narrowing treatment options. The addition of acquired azole resistance can often make combination therapies the only treatment option.

We suggest that drugs used in the clinical setting need to differ structurally from those used in agriculture, if they act via the same cellular mechanism. For example, should long-tailed azoles be reserved for medical applications and not developed as agrochemicals? The alternative is to focus on the development of separate drugs and agrochemicals that attack different cellular targets in order to overcome the cross-habitat resistance found throughout the fungal kingdom.

The knowledge gained from understanding how key AA changes in the ligand-binding pocket of SDMs affect substrate and inhibitor binding can now be used to design a new generation of azole drugs that are more potent and resistant to fungal adaptation via the known mechanisms. Stronger binding could be achieved by modulating key interactions of these ligands with the active site and the substrate entry channel of SDMs.

## Figures and Tables

**Figure 1 jof-07-00001-f001:**
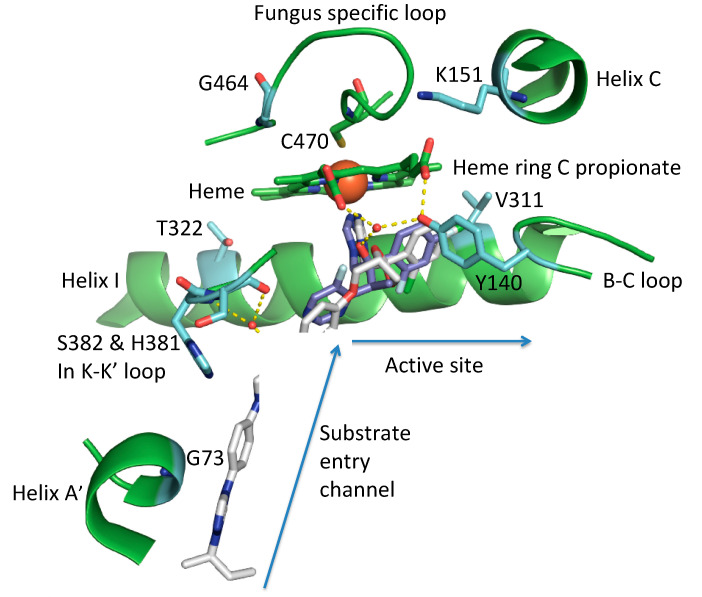
Structural view of the ligand-binding pocket of *S. cerevisiae* SDM (sterol 14-alpha demethylase) with the azole inhibitor short-tailed VCZ (voriconazole) and the long-tailed ITC (itraconazole) overlaid. The triazole of each drug is the 5th ligand of the heme iron (large red ball) and C470 is the 6th ligand. AA (amino acid) residues important for azole binding and/or azole resistance discussed in this review are shown with their carbons in light green. A list of this AAs is given in Appendix A. Relevant structural features, including the heme porphyrin, are given in green. Water-mediated hydrogen bond networks important for the binding of VCZ in the active site and ITC in the substrate entry channel are shown as yellow dashes.

**Figure 2 jof-07-00001-f002:**
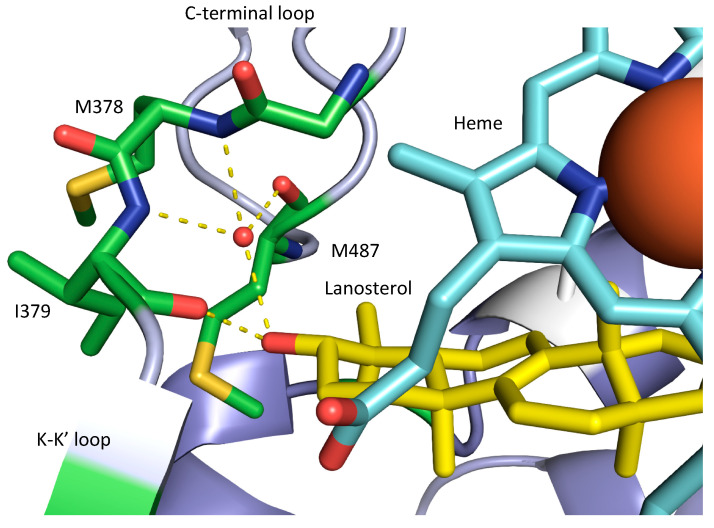
Structural view of the ligand-binding pocket of the human CYP51 D231A H314 mutant catalytic domain (Protein Data Bank ID: 6UEZ) interacting via hydrogen bonds (yellow dashes) with the hydroxyl of lanosterol (carbons in yellow). The heme is given in blue with the iron as a large red ball. Amino acid residues involved in the hydrogen bonding directly or as part of a water-mediated network are indicated. A list of all key AA residues for the main structures of the SDM are given in Appendix A.

**Table 1 jof-07-00001-t001:** Pathogenic fungi with naturally occurring amino acid (AA) substitutions in the active site of sterol 14α-demethylase (SDM; CYP51 enzyme of the cytochrome P450 superfamily) and correlated azole resistance.

	Genus	Species	Phylum	AA Change in Active Site	2nd Mutation	Gene	Correlated Resistance *	Ref.

**animal-pathogenic fungi**	*Ajellomyces*	*capsulatus*	Ascomycota	Y136F		*ERG11*	FLC, ITC	[28]
	*Aspergillus*	*fumigatus*	Ascomycota	Y121F	(TR46) T289A	*cyp51A*	VCZ	[31,32,33,34,35]
	*Aspergillus*	*fumigatus*	Ascomycota	I301		*cyp51A*	FLC	[36]
	*Aspergillus*	*lentulus*	Ascomycota	n.a.		*cyp51A*	VCZ	[37]
	*Candida*	*albicans*	Ascomycota	G464S		*ERG11*	FLC	[38]
	*Candida*	*albicans*	Ascomycota	Y132F, K143R		*ERG11*	FLC, VCZ	[39,40]
	*Candida*	*albicans*	Ascomycota	T315A		*ERG11*	FLC	[41]
	*Candida*	*auris*	Ascomycota	Y132F, K143R		*ERG11*	FLC	[41]
	*Candida*	*orthopsilosis*	Ascomycota	Y132F, K143R		*ERG11*	FLC, VCZ	[42]
	*Candida*	*parapsilosis*	Ascomycota	Y132F	P6S, C45G, G50L, D460T	*ERG11*	FLC, VCZ	[43,44]
	*Candida*	*parapsilosis*	Ascomycota	Y132F	R398I	*ERG11*	FLC, VCZ	[45,46]
	*Candida*	*tropicalis*	Ascomycota	Y132F, K143R		*ERG11*	FLC, VCZ	[40,43]
	*Cryptococcus*	*neoformans*	Basidiomycota	Y145F		*ERG11*	VCZ	[47]
	*Kluyveromyces*	*marxianus*	Ascomycota	K151		*ERG11*	FLC, VCZ, ITC, PCZ	[48]
	*Mucor*	*circinelloides*	Mucormycota	F129	A291	*CYP51 F5*	FLC, VCZ	[49]
	*Rhizopus*	*arrhizus*	Mucormycota	F129	A291	*CYP51 F5*	FLC, VCZ	[49]
	*Rhizopus*	*microsporus*	Mucormycota	F129	A291	*CYP51 F5*	FLC, VCZ	[49]
	*Scedosporium*	*apiospermum*	Ascomycota	Y136F		*cyp51*	VCZ	[50]
**plant-pathogenic fungi**	*Blumeria*	*graminis*	Ascomycota	Y136F/Y137F		*Cyp51*	TBC	[51]
	*Mycosphaerella*	*graminicola*	Ascomycota	Y137F, G460		*Cyp51*	TDM, TBC, EPC	[52,53]
	*Parastagonospora*	*nodorum*	Ascomycota	Y144F/H		*Cyp51*	PPC	[54]
	*Puccinia*	*triticina*	Basidiomycota	Y134F		*Cyp51*	EPC	[55]
	*Uncinula*	*necator*	Ascomycota	Y136F		*Cyp51*	TBC	[56]

Legend. Not available (n.a.), animal pathogenic fungi (includes fungi causing infections in animals and humans), gene (name of sterol 14α-demethylase-encoding gene carrying the AA substitution), fluconazole (FLC), itraconazole (ITC), posaconazole (PCZ), voriconazole (VCZ), tebuconazole (TBC), propiconazole (PPC), epoxiconazole (EPC), triadimenol (TDM), 46bp-tandem repeat in the promotor region of the 14α-demethylase gene (TR), quality of wild-type AA (quality of wt), position and quality of the amino acid change (AA change) amino acid (AA), phenylalanine (F), tyrosine (Y), threonine (T), isoleucine (I), arginine (R), lysine (K), cysteine (C), leucine (L), alanine (A), aspartic acid (D), histidine (H), proline (P), serine (S), *Ajellomyces capsulatus* (=*Histoplasma capsulatum*). The exact amino acid exchanges in *A. lentulus* are not described yet. * Resistance calling was performed according to EUCAST breakpoints (see Appendix A). Key AAs are given in Appendix A.

**Table 2 jof-07-00001-t002:** Relevant amino acids (AAs) in the ligand-binding pocket of sterol 14α-demethylase based on *Saccharomyces cerevisiae* lanosterol 14α-demethylase (see Figure 1 and Appendix A). An overview of mutations in structurally aligned amino acid position in other pathogenic fungi of plants and animals is given below.

*Saccharomyces*	*Cerevisiae*	Y140	T322	G464	K151	G73	V311
*Ajellomyces*	*capsulatus*	Y136F	-	-	-	-	-
*Aspergillus*	*fumigatus*	Y121F	I301	-	-	G54	A289
*Aspergillus*	*lentulus*	-	-	G448S	-	-	-
*Candida*	*albicans*	Y132F	T315A	G464S	K143R	-	-
*Candida*	*auris*	Y132F	-	-	K143R	-	-
*Candida*	*orthopsilosis*	Y132F	-	-	K143R	-	-
*Candida*	*parapsilosis*	Y132F	-	-	-	-	-
*Candida*	*tropicalis*	Y132F	-	-	K143R	-	-
*Cryptococcus*	*neoformans*	Y145F	-	-	-	-	-
*Kluyveromyces*	*marxianus*	-	-	-	K151	-	-
*Mucor*	*circinelloides*	F129	-	-	-	-	A291
*Rhizopus*	*arrhizus*	F129	-	-	-	-	A291
*Rhizopus*	*microsporus*	F129	-	-	-	-	A291
*Scedosporium*	*apiospermum*	Y136F	-	G464S	-	-	-
*Blumeria*	*graminis*	Y136F	-	-	-	-	-
*Mycosphaerella*	*graminicola*	Y137F	-	G460	-	-	-
*Parastagonospora*	*nodorum*	Y144F/H	-	-	-	-	-
*Puccinia*	*triticina*	Y134F	-	-	-	-	-
*Uncinula*	*necator*	Y136F	-	-	-	-	-

Legend Phenylalanine (F), tyrosine (Y), threonine (T), isoleucine (I), arginine (R), lysine (K) phenylalanine (F), tyrosine (Y), threonine (T), isoleucine (I), arginine (R), lysine (K), alanine (A), histidine (H), serine (S), valine (V), amino acid substitution not yet described as relevant in these fungi (-), *Ajellomyces capsulatus* (=*Histoplasma capsulatum*).

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
