# Peer review of "Sterol 14α-Demethylase Ligand-Binding Pocket-Mediated Acquired and Intrinsic Azole Resistance in Fungal Pathogens"

_jof, 2020, doi:10.3390/jof7010001_

Round 1
Reviewer 1 Report
The paper is very interesting as a review being focused on one of the mechanisms involved in fungal azole resistance. There are mentioned three mechanisms but it is deeply analyzed one of them: AA substitutions with the ligand-binding pocket of SDMs.
As a result of binding azoles to the active site of SDM, takes place the depletion of ergosterol, the accumulation of toxic compounds and finally the inhibition of growth due to low membrane fluidity and integrity lipid layer.
I recommend the authors to explain clearly at the beginning of the article the genetic support of azole resistance. Some information about Stachybotrys is needed.
If K.R. is a funded PhD student by the FWF grant P32329-B entitled ‘Intrinsic azole resistance in mucormycetes I suggest adding some experimental results
Author Response
Reviewer 1
The paper is very interesting as a review being focused on one of the mechanisms involved in fungal azole resistance. There are mentioned three mechanisms but it is deeply analyzed one of them: AA substitutions with the ligand-binding pocket of SDMs.
As a result of binding azoles to the active site of SDM, takes place the depletion of ergosterol, the accumulation of toxic compounds and finally the inhibition of growth due to low membrane fluidity and integrity lipid layer.
I recommend the authors to explain clearly at the beginning of the article the genetic support of azole resistance. Some information about Stachybotrys is needed.
Response: The title of the paper and the introduction announces that the article will be solely focused on the AA substitution of the ligand-binding pocket of SDMs. The comment of the reviewer is to general and we therefore do not know how to address it. What do you specifically mean with the genetic support that should be included? We were unable to track down publications that descript AA substitutions in the ligand-binding pocket of the SDMs of Stachybotrys. Please can you provide us with references that you had in mind to be included.
If K.R. is a funded PhD student by the FWF grant P32329-B entitled ‘Intrinsic azole resistance in mucormycetes I suggest adding some experimental results.
Response: This is a pure review article and it was appointed as such with the editor. The PhD student drafting the article is paid by the FWF grant therefore this is given as funding source. We do not have the intention to publish original data in this review article. Data will be published in a research article when we do have enough for a story.
Reviewer 2 Report
This manuscript is a well written and thorough review of amino acid substitution mutations leading to azole resistance in fungi pathogenic for humans and/or plants. The literature review appears quite comprehensive. Significance and relevance are explained well, i.e., the growing occurrence of such mutations, the medical and agricultural consequences, and the need for appropriate antibiotic stewardship in agriculture and medicine.
In general, the medical and agricultural background of the pathogens is accurate, with one exception. Lines 337-338 contain the statement, “Cryptococcus is dimorphic and grows as a filamentous fungus in the environment.” This description is accurate for Histoplasma capsulatum but not for Cryptococcus. Cryptococcus can form filaments as part of its mating process, but the asexual vegetative form grows as a yeast in the environment and the host, and it does make a capsule, which is more prominent in the host environment. The yeast morphology should be described for Cryptococcus, and it might be useful to move the dimorphism description to the text on Histoplasma (lines 384-392, which currently lacks any morphological description; note that Histoplasma capsulatum does not have a capsule, despite its name.
Other suggestions are largely typographical or stylistic.
1) Why are AMB & IVU abbreviations provided in the Table 1 legend when they do not appear in the table itself?
2) Is table S2 meant not to have genus designations?
3) Tables S2 and S3 have AmB in table and AMB in legend.
4) The 2nd Table S2 is probably meant to be Table S4.
5) Line 139 refers to Histoplasma capsulatum while Tables 1 and 2 show Ajellomyces capsulatus. Their identity relationship is explained later in the text (ll 385-386), but at this first occurrence, the name Ajellomyces capsulatus should probably be used to match the tables, as it is on line 192.
Author Response
Reviewer 2
This manuscript is a well written and thorough review of amino acid substitution mutations leading to azole resistance in fungi pathogenic for humans and/or plants. The literature review appears quite comprehensive. Significance and relevance are explained well, i.e., the growing occurrence of such mutations, the medical and agricultural consequences, and the need for appropriate antibiotic stewardship in agriculture and medicine.
Response: Thank you
In general, the medical and agricultural background of the pathogens is accurate, with one exception. Lines 337-338 contain the statement, “Cryptococcus is dimorphic and grows as a filamentous fungus in the environment.” This description is accurate for Histoplasma capsulatum but not for Cryptococcus. Cryptococcus can form filaments as part of its mating process, but the asexual vegetative form grows as a yeast in the environment and the host, and it does make a capsule, which is more prominent in the host environment. The yeast morphology should be described for Cryptococcus, and it might be useful to move the dimorphism description to the text on Histoplasma (lines 384-392, which currently lacks any morphological description; note that Histoplasma capsulatum does not have a capsule, despite its name.
Response: Thank you for this comment. We changed this paragraphs according to your suggestions.
Other suggestions are largely typographical or stylistic.
- Why are AMB & IVU abbreviations provided in the Table 1 legend when they do not appear in the table itself?
Response: This was a copy paste error and we corrected it.
2) Is table S2 meant not to have genus designations?
Response: We corrected that.
3) Tables S2 and S3 have AmB in table and AMB in legend.
Response: We corrected that.
4) The 2nd Table S2 is probably meant to be Table S4.
Response: We corrected that.
5) Line 139 refers to Histoplasma capsulatum while Tables 1 and 2 show Ajellomyces capsulatus. Their identity relationship is explained later in the text (ll 385-386), but at this first occurrence, the name Ajellomyces capsulatus should probably be used to match the tables, as it is on line 192.
Response: We changed that both names are now given at the first occurrence in the text. We also added this information in the table legend.